# Enhancing the Sensory Quality, Stability, and Shelf Life of Baobab Fruit Pulp Drinks: The Role of Hydrocolloids

**DOI:** 10.3390/polym17101396

**Published:** 2025-05-19

**Authors:** Abdullahi Idris Muhammad, Abdulrashid Rilwan, Zahrau Bamalli Nouruddeen, Ovinuchi Ejiohuo, Nasser Al-Habsi

**Affiliations:** 1Department of Food Science and Technology, Kano University of Science and Technology, Wudil, Kano 713101, Nigeria; 2Department of Food Science and Nutrition, Sultan Qaboos University, Al-Khoudh P.O. Box 34, Muscat 123, Oman; 3Doctoral School, Poznan University of Medical Sciences, Bukowska 70, 60-812 Poznan, Poland

**Keywords:** baobab fruit pulp, functional drinks, hydrocolloids, xanthan gum, carboxy methylcellulose, shelf life, viscosity, stability, consumer acceptance

## Abstract

Baobab (*Adansonia digitata* L.) fruit pulp (BFP) is particularly noted for its high concentrations of bioactive compounds, including polyphenols, vitamins (notably vitamin C), and dietary fiber, surpassing common fruits such as oranges in ascorbic acid content. Despite its long-standing use in local communities as a functional food ingredient, BFP drinks face significant challenges related to their sensory parameters and shelf life, particularly due to rapid microbial growth under tropical conditions. This study investigated the effects of two hydrocolloids, xanthan gum (XG) and carboxymethyl cellulose (CMC), on the viscosity, shelf-life stability, and consumer acceptance of BFP drinks. Seven samples were formulated with these hydrocolloids at different concentrations, namely, BXG1 (95% BFP:5% XG), BXG2 (90% BFP:10% XG), BXG3 (85% BFP:15% XG), BCMC1 (95% BFP:5% CMC), BCMC2 (90% BFP:10% CMC), and BCMC3 (85% BFP:15% CMC), alongside a control sample (100% BFP) and a commercially synthetic drink (CSD) for comparison. The results indicate that BFP drink sample (BXG1) stored under refrigeration (4 °C) for up to 14 days retains acceptable sensory qualities with minimal microbial growth (9 CFU/mL). However, storing at room temperature (ca. 25 ± 2 °C) led to rapid microbial proliferation (oral observation) within four days. These findings also confirm that BFP drinks can provide significant nutritional value, offering 330.64 kcal/100 g of metabolizable energy. This study suggests that, while BFP drinks offer several healthy benefits, enhancing their stability using hydrocolloids and appropriate storage conditions is essential. Future studies should focus on the incorporation of natural preservatives to enhance their stability while preserving their nutritional integrity.

## 1. Introduction

Africa is renowned for its abundant indigenous plant species, rich in health-promoting compounds, yet many remain underutilized [1,2]. Among these is the baobab (*Adansonia digitata* L.) tree (Figure 1), found predominantly in Africa and Australia and includes eight species [3]. Six of these species are endemic to Madagascar [4], with its most prevalent species, *Adansonia digitata* L., thriving in the savannas of sub-Saharan Africa and western Madagascar [2,5,6]. The dispersal of baobab species across continents is attributed to historical trade routes facilitated by French, Portuguese, and Muslim traders, resulting in its presence in countries such as Cuba, Haiti, Indonesia, Malaysia, Mauritius, Mozambique, the Netherlands, Saudi Arabia, Somalia, Sudan, Tanzania, the USA, Oman, and Yemen, as well as on islands, such as Virgin and Zanzibar [3,6,7]. The baobab fruit has a capsule shell, is ellipsoid, ovoid, 20–30 cm long, and up to 10 cm in diameter, which is covered on the outside with greenish-brown felted trichomes [3]. The shell contains numerous hard, brownish seeds, rounded or ovoid, up to 15 mm long (Figure 2A), which are embedded in a yellowish-white, floury acidic (ca. pH 3.2) pulp (Figure 2B) that has a slight lemony flavor. The brown seeds are arranged in rows with 2–8 locules per fruit [3,8]. The seeds are attached to fibrous strands extending from the wall of the fruit [9].

Baobab fruit pulp (BFP) has garnered increasing interest due to its nutritional richness as it contains naturally occurring bioactive compounds that can be a significant contributor to the daily intake of important nutrient and non-nutrient compounds [9,10]. Studies have highlighted the fruit’s richness in essential nutrients, such as polyphenols, minerals, and vitamins, particularly vitamin C, surpassing concentrations typically found in common fruits. For example, BFP is 10-times higher in ascorbic acid than that of orange [11,12] and also contains a high amount of both soluble and insoluble dietary fiber [13,14,15,16]. Vitamin C is a well-studied antioxidant that aids in absorbing non-heme iron in the mammalian body. By converting non-heme iron into a more absorbable form, ascorbic acid enhances iron uptake from plant-based foods, which can contribute to physiological iron levels. The antioxidant properties of vitamin C also help protect cells from oxidative damage caused by free radicals and other reactive oxygen species, and its role supports overall nutritional health [2,17]. The high potassium content of BFP assists in regulating blood pressure, while its minimal sodium content also promotes good heart health [18]. BFP also contains appreciable quantities of other essential vitamins, such as B1, B2, and B3, further enhancing its nutritional profile [19,20,21].

Notably, other studies reported that the bio-accessible polyphenols present in BFP have shown potential in reducing the glycemic response to carbohydrate-rich foods both in vivo [4] and in vitro [22], further underscoring its health benefits. Commercial interest in BFP increased after the EU authorized “baobab pulp” as safe for human consumption under decision 2008/575/EC of the European Parliament [23]. However, since before this legislation, BFP has been of significant importance and primarily consumed by local African communities, either by sucking the diluted liquid extract from the pulp [11] or by adding it to thick grain preparations to make thinner gruels, sour dough, a coffee-like drink, or dried for future utilization [3,24]. A study conducted by Igboeli et al. revealed that in the northern part of Nigeria and some other West African locales, a refreshing drink is made from BFP and cold water, which is a known practice to exploit the nutritional benefits of fruit extract, such as BFP [25]. Also, Sidibe et al. confirmed that the cattle-owning Fulani and the Hausa peoples of northern Nigeria consume a drink composed of a BFP emulsion mixed with milk, producing a pleasing wine gum flavor [3].

Despite the well-documented nutritional potential of BFP-based drinks, their widespread consumption is hindered by significant stability and safety concerns. The limited shelf life of these beverages, exacerbated by tropical storage conditions and the lack of advanced food processing and hygienic preservation methods, greatly compromises their quality and consumer safety. The high carbohydrate content inherent to BFP creates an ideal environment for microbial proliferation. In the absence of adequate preservation, fermentation typically begins in the range of 24–48 h post-production under ambient tropical conditions. This results in the accumulation of alcohol and carbon dioxide, leading to undesirable carbonation, off-flavor development, and the formation of mucilaginous precipitates. Sedimentation, a physicochemical issue previously attributed to the intrinsic properties of the fruit, further exacerbates the drink’s instability [9]. Alarmingly, despite visible spoilage, anecdotal reports revealed that some consumers, driven by their strong preference for BFP-derived beverages, continue to ingest these compromised BFP-based drinks, thereby exposing themselves to foodborne illnesses, particularly gastrointestinal disturbances. This not only undermines the health-promoting attributes of BFP but also points to a larger systemic issue of the limited research attention given to indigenous and underutilized fruits and their derivatives. Therefore, further investigation is urgently warranted to develop innovative preservation strategies that can stabilize BFP drinks. Such advancements are critical to maximizing the safe utilization of these nutritionally rich beverages, while simultaneously contributing to food security and enhancing dietary diversity, especially among resource-poor populations. This underscores the urgent relevance of the present study in addressing both public health risks and broader food system challenges.

At present, there is a notable scarcity of studies focused on enhancing techno-functionality of the BFP-based beverages using hydrocolloids. These hydrocolloids are extensively utilized in the food industry due to their multifunctional properties, including thickening, gelling, encapsulating, and swelling. Compared to synthetic and semi-synthetic alternatives, they offer distinct advantages, such as enhanced stability, biocompatibility, non-toxicity, biodegradability, sustainability, renewability, broad availability, cost-effectiveness, and chemical modifiability [26,27,28]. These characteristics make them especially valuable in improving the quality of fruit-based products, aligning with global trends for healthier and more sustainable food choices [26,27,28,29,30]. Among these hydrocolloids, xanthan gum (XG) and carboxymethyl cellulose (CMC) are among the most commonly employed in food formulations. Both share a cellulose backbone composed of β-(1,4)-D-glucose units, but differ in their side chains: XG contains a trisaccharide side chain, while CMC incorporates carboxymethyl groups [28,31]. Numerous studies have demonstrated the effectiveness of these hydrocolloids in enhancing the functional and sensory properties of various fruit-based beverages, such as in red dragon fruit drinks [32], raspberry juice [33], soy–moringa beverages [34], longan juice [35], etc.

To the best of our knowledge, no research has investigated the effects of any hydrocolloid incorporation on the viscosity, physicochemical stability, and sensory properties of BFP drinks. In response to this gap, this study explores the use of these two widely recognized hydrocolloids; XG and CMC as multifunctional agents aimed at enhancing the stability, viscosity, and sensory attributes (appearance, taste, mouthfeel, flavor, and general acceptability) of the developed BFP–hydrocolloid beverages. The primary objective of this study is to evaluate the individual impact of XG and CMC on the rheological and sensory characteristics, as well as the shelf-life stability of these drinks. Microbial growth was also assessed on days 1 and 14 under both refrigerated (4 °C) and ambient storage conditions, using total mesophilic aerobic plate counts to determine the hygienic quality of the product over time. The findings from this research will provide foundational insights into the formulation of safer, more stable, and consumer-acceptable BFP-based beverages.

## 2. Materials and Methods

### 2.1. Chemicals and Reagents

All chemicals were obtained from Sigma Aldrich, Chemie GmbH (Taufkirchen, Germany) unless noted otherwise. All the glassware used in this study was procured from Glassco Laboratory Equipment Pvt. Ltd. (Haryana, India), and Pyrex, (Corning, NY, USA).

### 2.2. Materials Collection

Granulated, fresh BFP (Figure 2B) was obtained from Rimi Central Market (Kano, Kano State, Nigeria). The hydrocolloids used were both food grade. Xanthan gum (Freee^TM^, all-in-one gluten-free, produced by Doves Farm Foods Ltd., (Hungerford, Berkshire, UK) was purchased from Lulu Hypermarket (Al-Gharrafa, Doha, Qatar). Carboxymethyl cellulose was produced and purchased from Grace Co. Ltd. (Ifako-Ijaiye, Lagos, Nigeria).

### 2.3. Sample Preparation

BFP powder was prepared by gently pounding the seed granules with a mortar and pestle to remove the powder from the seeds, sieved, and blended into a fine powder (Figure 2C). The BFP powder was stored at room temperature (ca. 25 ± 2 °C) until formulation (ca. 2 h) for further juice extraction. Potable but non-sterile drinking water was used throughout the cleaning and sample operation to mimic typical preparation practices.

### 2.4. Extraction and Thickening of Baobab Fruit Pulp Drink

BFP juice was extracted according to the methods reported by Tembo et al. [17] and Adedokun et al. [36], with slight modifications. A cumulative measure of 50 g of the fine-powdered BFP, after deducting respective amounts of hydrocolloids (XG/CMC), was then added into 1.5 L of pre-boiled and cooled drinking water, homogenized, filtered through one layer of muslin cloth, and stored at room temperature (ca. 25 ± 2 °C) prior to mixing with each respective hydrocolloid, which were also diluted with 500 mL of pre-boiled and cooled water. XG and CMC were added to the BFP drink at a concentration of 0, 5, 10, and 15%, as coded in Figure 3. All dilutions were carried out in clean bowls, and refined white sugar was added to a separate batch, mainly for sensory evaluation, as Adedayo et al. [12] suggested. The formulated drinks were packaged in 50 mL disposable polyethylene bottles, labeled, and immediately stored at refrigeration temperature (4 °C) for subsequent analyses. No artificial colors or flavors were included in the formulations. Figure 4 presents a flow diagram of the unit operations involved in the juice extraction and thickening of the BFP drink. A Commercial Synthetic Drink (CSD), which was considered the most liked by local consumers in the Kano State of Nigeria, was purchased from a local canteen in Wudil, Nigeria, and served as a comparative reference control in this study and for the evaluation of consumer acceptability. The formulated drinks were then subjected to analyses to determine their makeup and related characteristics.

### 2.5. Physicochemical and Functional Properties

#### 2.5.1. Titratable Acidity (TA) and pH Measurements

The titratable acidity (TA) of the prepared BFP drink sample was determined according to the AOAC 942.15 titrimetric method, as described by Saldanha et al. 2025 [37]. An aliquot of 10 mL was titrated with a standardized 0.1 N NaOH solution to the phenolphthalein endpoint. TA was expressed as grams of ascorbic acid per 100 mL of sample and calculated using the formula:(1)$$TA\;(g/100\;mL)=\frac{V\times N\times EqWt}{V_{s}}\times 100$$
where *V* is the volume of NaOH used (mL), *N* is the normality of NaOH, *EqWt* is the equivalent weight of ascorbic acid (88.06 g/mol), and vs. is the volume of the sample titrated (mL).

The pH of the sample was measured following the method described by Tembo et al. (2017) [17]. A 10% (*w*/*v*) suspension of the sample was prepared by mixing the sample with deionized water that had been previously boiled and cooled to room temperature (ca. 25 ± 2 °C). The mixture was homogenized using a single-speed waring micro-blender (Thomas, Scientific, Swedesboro, NJ 08085, USA) and transferred into a 250 mL beaker. The pH was measured directly after equilibration at room temperature (ca. 25 ± 2 °C) using a calibrated handheld pH meter (Hanna Instruments, Model HI2211, Bedfordshire, UK).

#### 2.5.2. Total Soluble Solids (TSSs)

Total soluble solids (TSSs) were analyzed using a Hanna HI96801 Digital Brix Refractometer 0–85% Maple SAP (Syrup National Industrial Supply, Temecula, CA, USA).

#### 2.5.3. Bulk Density

The bulk density was determined by weighing a 10 mL volume of a BFP powdered sample in a graduated cylinder that had been compacted by tapping on the benchtop several times until there was no further diminution of the sample level after filling to the 10 mL mark, as described in AOAC, 1996 [38], and was calculated using the formula:(2)$$Bulk\;density(g/mL)=\frac{weight\;of\;sample(g)}{volume\;of\;sample(mL)}$$

#### 2.5.4. Water Absorption Capacity (WAC)

The water absorption capacity of the sample was determined using the method described in [39] with slight modifications. A 2 g sample was weighed into a pre-weighed centrifuge tube, and 20 mL of distilled water was added. The mixture was vortexed for 30 s to ensure thorough hydration and then allowed to stand at room temperature (ca. 25 ± 2 °C) for 30 min. Following equilibration, the mixture was centrifuged at 3000× *g* in a fixed-angle (6 × 50 mL) rotor for 15 min using a universal refrigerated microprocessor-controlled table-top centrifuge (SIGMA 2-16PK, Sigma Laboratory Centrifuges, 37520 Osterode am Harz, Germany). The supernatant was decanted, and the residue was weighed. The water absorption capacity was calculated using the following formula, and the results are expressed in grams of water absorbed per gram of sample:(3)$$WAC(g/g)=\frac{weight\;of\;hydrated\;residue\left(g\right)-weight\;of\;dry\;sample\;(g)}{weight\;of\;dry\;sample\;(g)}$$

#### 2.5.5. Solubility and Swelling Power (SP)

The solubility and swelling power of the sample were determined using a modified method of Leach et al. (1959), as described in [39]. A 2 g sample was suspended in 20 mL of distilled water in a centrifuge tube. The suspension was heated at 90 °C for 30 min in a water bath with constant stirring to ensure uniform heating. After heating, the suspension was centrifuged at 3000× *g* for 15 min. The supernatant was decanted into a pre-weighed beaker and dried at 105 °C to a constant weight. The weight of the dried residue was recorded as soluble material. The swollen residue in the centrifuge tube was weighed to determine the swelling power. The results were calculated using the formula:(4)$$Solubility(\%)=(\frac{weight\;of\;dried\;soluble\;material\;\left(g\right)}{weight\;of\;dry\;sample\;\left(g\right)})\times 100$$(5)$$Swelling\;power\;(g/g)=\frac{weight\;of\;swollen\;residue\;(g)}{weight\;of\;dry\;sample\;\left(g\right)}$$

#### 2.5.6. Vitamin C Content

The vitamin C content of the BFP thickened drink samples was determined using the method recommended by Nielsen [40]. Briefly, 25 mL of sample was weighed into 100 mL volumetric flasks in triplicate. Next, 25 mL of 20% metaphosphoric (0.5% oxalic acid) was added as a stabilizing agent and diluted to a 100 mL volume. Ten mL was then pipetted into a 250 mL flask into which 2.5 mL of acetone was added. It was then titrated using an indophenol solution (2,6-dichlorophenolindophenol) to a faint pink color, which persisted for 15 s.

#### 2.5.7. Viscosity

Viscosity was measured using a viscometer (Brook field DV-II+, Brookfield Engineering Laboratories Inc., Middleboro, MA, USA) with a temperature control, sample adaptor (No. 13), and spindle (SC418) as described in the IUPAC methods (2000). The samples were poured into the sample adaptor, and the spindle, attached to the viscometer, was immersed into the sample. The viscosity of the extract was measured at a 30 RPM rotor speed and the extract temperature was maintained at 30 °C. The sample viscosity was measured in centipoises (cP).

### 2.6. Microbial Analysis

The formulated drink was examined for microbiological quality using the total mesophilic plate count method [41]. Briefly, 1 mL of the drink, following ten-fold serial dilutions (i.e., 10^−1^, 10^−2^, and 10^−3^), was aseptically aliquoted onto disposable, sterile, total aerobic Petri plates containing nutrient agar (ca. 13 mL) in a sterile environment. This was followed by incubating at 37 °C for 18 h. The total mesophilic aerobic plate count was expressed as log CFU/mL of sample.

### 2.7. Sensory Evaluation

A preliminary acceptability test was conducted using 20 panelists consisting of students and laboratory staff (aged between 22 and 35 years, 60% female, all with prior exposure to BFP-based products) from the Food Processing Laboratory of the Department of Food Science and Technology, Kano University of Science and Technology, Wudil. A 7-point hedonic scale was used to access five sensory attributes: taste, mouthfeel, flavor, color, and overall acceptability, where 1 = Dislike very much and 7 = like very much. The test was conducted on the final formulated BFP drinks (*n* = 7) as well as on the CSD control in comparison with control samples. The same panelists were used throughout the entire study period to ensure consistency. Water was provided to the panelists to clear and rinse their palate after tasting each sample.

### 2.8. Proximate Composition

The proximate composition of the BFP was analyzed according to the AOAC standard methods as follows:

#### 2.8.1. Moisture Content

A clean, dried, glass Petri dish was weighed and marked as W_1_. Five grams of the sample was placed onto the glass Petri dish and weighed as *W*_2_. This was then oven-dried in the glass Petri dish at 150 °C for 30 min. The dried sample was then cooled in a desiccator for 10–15 min and then weighed as *W*_3_ [41]. The moisture content was calculated as:(6)$$Moisture\;Content\left(\%\right)=\frac{W_{2}-W_{3}}{W_{2}-W_{1}}\times 100$$

#### 2.8.2. Ash Content

The weight of a clean, dried crucible was marked as *W*_1_. Five grams of the BFP was placed into the crucible and weighed as *W*_2_. Ashing was conducted in a muffle furnace at 550 °C for 18 h, removed from the muffle furnace, cooled in a desiccator for 45 min, and weighed as *W*_3_ [41].(7)$$Ash\;Content\left(\%\right)=\frac{W_{3}-W_{1}}{W_{2}-W_{1}}\times 100$$

#### 2.8.3. Crude Lipid Content

Filter paper was weighed as *W*_1_, and two grams of sample was weighed as *W*_2_ and placed in a cellulose extraction thimble (Whatman, 33 mm × 80 mm). Next, 250 mL of petroleum ether was added to a round-bottom flask on top of a heating mantle heated to 70 °C, and water was circulated in the condenser. The extraction was conducted for ca. 6 h, after which time, the sample was removed and the thimble was allowed to drain and air-dried. It was then weighed again and recorded as *W*_3_ [38].(8)$$Crude\;Lipid\left(\%\right)=\frac{W_{2}-W_{3}}{W_{2}}\times 100$$

#### 2.8.4. Crude Protein Content

Total protein content was determined using the Kjeldahl method. Briefly, a sample of BFP (ca. 0.2 g) was weighed and transferred onto filter paper and then into the Kjeldahl flask, along with 15 mL of concentrated sulfuric acid (H_2_SO_4_) and 1 tablet of Kjeldahl catalyst. The flask was heated gently in an inclined position in a fume cupboard using a heating mantle, and the flask was swirled intermittently. When the initial vigorous reaction was partially cooled, the heat was increased and digestion continued until the solution became colorless. The flask continued to be swirled intermittently to wash charred particles from the sides of the flask.

The flask was cooled down and the contents transferred into a 100 mL volumetric flask, diluted to the 100 mL marked level with distilled water, and then 10 mL of BFP + 15 mL of 40% NaOH was transferred into the distillation apparatus consisting of the 500 mL flask, topper carrying a dropping funnel, and a splash head adaptor (a vertical condenser, attached to a straight delivery tube). Next, 10 mL of 2% boric acid solution was added to a 250 mL conical flask and a few drops of screened methyl red indicator were added to the flask and placed on the receiver. A few pieces of granulated zinc and some anti-bumping granules were added to the distillation flask, and the apparatus was shaken gently to ensure maximum mixing of the solution. The flask was then boiled vigorously until ca. 25 mL was distilled, and then the receiver was removed and titrated with 0.025 M H_2_SO_4_ to the pink-color end point (titer value).(9)$$Nitrogen\left(\%\right)=\frac{\left(T-B\right)\times N\times 1.4007}{W}$$
where *T* is the titration volume of the sample (mL), while *B* is for the blank (mL), *N* is the normality of the sulfuric acid, and *W* is the weight of sample (g).(10)$$Protein\left(\%\right)=Nitrogen\;(\%)\times 6.25$$

#### 2.8.5. Crude Fiber Content

Five grams of BFP powder was weighed (*W*_1_) and defatted by ether extraction with a Soxhlet apparatus, dried, and transferred into a 600 mL beaker of the fiber digestion apparatus. Next, 400 mL of 1.25% sulfuric acid was added to the beaker. The beaker was placed on a digestion apparatus with a pre-adjusted heater, boiled for 30 min, removed, and the contents were filtered using a muslin cloth. The beaker was then rinsed with 25 mL of 1.25% H_2_SO_4_ and washed through the muslin cloth. The residue was returned to the beaker, 400 mL of 1.25% NaOH was added to the beaker, boiled for 30 min, and then the beaker was removed and filtered as described earlier. The residue was then washed with 25 mL of 1.25% NaOH, the fiber mat and the residue were oven-dried at 150 ± 2 °C for 30 min, cooled in a desiccator, and weighed as *W*_2_. Next, the sample was ignited in a muffle furnace at 500 °C ± 15 °C for 30 min, the dish was removed, cooled in a desiccator, and weighed as *W*_3_ [38].(11)$$Fibre\left(\%\right)=\frac{W_{2}-W_{3}}{W_{1}}\times 100$$

### 2.9. Statistical Analysis

Statistical analysis was performed using Statistical Package for the Social Sciences (SPSS) software, version 20, to analyze the results using one-way analysis of variance (ANOVA), and statistical significance was set at *p* < 0.05. The results are presented in the tables, graphs, and charts with mean values and standard deviations using Microsoft Package 2021.

## 3. Results and Discussion

### 3.1. Proximate Composition of the Powdered BFP

The quality of raw materials significantly influences the overall product quality [42] and dictates the after-harvest handling, processing, and preservation methodologies, which need to be implemented. This necessitates determining the proximate composition of the powdered BFP used in this study. The results reveal that the moisture content, crude fat, crude protein, ash, crude fiber, and carbohydrates are 10.02 ± 0.07, 4.09 ± 0.21, 7.13 ± 0.19, 4.82 ± 0.52, 7.50 ± 0.52, and 66.53 ± 0.52%, respectively. The carbohydrate content of the studied BFP was determined using the arithmetic difference method. Although the carbohydrate content was high (66.53%), it was less than the 76.2% reported by Osman [18], or 79.5% by Nour et al. [9]. Tembo et al. [17] reported that the bioactive and other components of the BFP depend on numerous factors, such as climatic conditions, plant variety, ripening stage of the plant, and harvesting time, thus supporting the disparate results reported in the proximate composition of BFP in various studies. Additionally, Assogbadio et al. [43] reported that genetic differences exist between baobab populations from different climatic provenances. Moreover, the metabolizable energy of the sample was calculated using the Atwater general factors, 4 kcal/g for protein, 9 kcal/g for fat, and 4 kcal/g for carbohydrate, resulting in metabolizable energy of 330.64 kcal/100 g, which is marginally higher than the 320.3 kcal/100 g reported by Osman [18]. This increased caloric value suggests the potential of BFP drinks as more nutrient-dense beverages, offering enhanced nutritional benefits and a valuable dietary option. This is particularly relevant in comparison to imitation nutritional drinks that have become prevalent in the market, especially in regions where affordable, nutrient-rich alternatives are scarce.

### 3.2. Functional Properties of the Studied BFP

The results of the BFP functional properties, including water absorption capacity (1.7 ± 0.07 *w*/*w*), solubility (0.44 ± 0.29%), swelling power (1.64 ± 0.42%), and bulk density (1.61 ± 0.52 *w*/*v*), display values that vary from those published by Adejuyitan et al. [44], who reported 0.92 *w*/*w* water absorption capacity, 1.10% solubility, 1.16% swelling power, and 0.4 *w*/*v* bulk density. Water absorption capacity is known to vary, depending on many factors, such as fruit ripeness, processing methods, and storage conditions. One among many possible explanations for the early spoiling of the formulated drinks could be the increased water absorption capacity of the BFP that was investigated in this work. This is evidenced from the work reported by Adejuyitan et al. [44], where water absorption capacity was used as a quality control measure to ensure consistency in the moisture content of baobab fruit products, which is crucial for maintaining the drink’s quality and extending shelf life.

### 3.3. Physicochemical Properties and Vitamin C Content

Table 1 displays the physicochemical characteristics and vitamin C content of the BFP drink, with varying hydrocolloid concentrations, as well as that of control samples.

The pH values of the formulated samples are in the range of 3.10–3.95, which are in the 2.5–4.0 range, considered satisfactory limits for acidified beverage preservation, as described by Woodroof and Phillips [45], and is similar to the one obtained by Tembo et al. [17]. The ANOVA of the samples containing hydrocolloids in different proportions shows a significant effect (*p* < 0.05) of the hydrocolloids on the pH with a dramatic increase. The pH value (3.10) obtained in the BFP-coded sample of this study is very comparable to the pH of 3.12 reported by Adedokun et al. [36]. This acidic pH value of the BFP-coded sample may be attributed to the high concentration of organic acid present, predominantly ascorbic acid, as reported in the research carried out by Tembo et al. [17]. Nwachukwu and Ezeigbo reported the significance of pH as a gauge for the microbial stability of foods, particularly in fruit juices, where a lower pH corresponds to greater microbiologic stability [46]. However, there was a significant increase (*p* < 0.05) in the pH level of drinks thickened at different concentrations, with sample BXG3 being the highest, while there was no significant difference (*p* < 0.05) between samples BXG1, BXG2, BXG3, BCMC1, BCMC2, and BCMC3 (Table 1). Based on this fact, it can be concluded that the addition of either of the hydrocolloids (XG and CMC) can reduce drink acidity, hence increasing the pH.

Sample CSD displayed the highest pH value at 4.20. Despite the relatively high pH, such drinks usually have a shelf life up to 6 months. This is likely due to the presence of preservatives, such as potassium sorbate, and the buffering capacity of citric acid, sodium citrate, and lactic acid in the ingredients, which helps maintain microbial stability. In contrast, the pH of the formulated BFP drink samples is in agreement with the pH of BFP reported in the published literature [17,47,48]. This pH range is also below the critical threshold of 4.6 for fruit juices, at which most of the spoilage microorganisms and pathogenic bacteria, such as *Clostridium botulinum*, cannot grow [49]; however, this does not guarantee complete safety without proper optimization of processing and preservation methods [48]. This necessitates that a detailed microbial profile of the thickened BFP drink should be conducted to assess whether its lower pH affects the microbial quality and safety during extended storage.

The titratable acidity (TA) of the samples was in the range of 2.68–4.18 (Table 1). The BFP-coded sample had the lowest pH of 3.10 and a TA value of 4.18. The expectation is that the increasingly lower pH values should correspond to higher acidity [50], which agrees with the results reported here. The predominant organic acid (ascorbic acid) present in baobab may be responsible for this high TA value as reflected in samples BXG1, BXG2, BXG3, BCMC1, BCMC2, and BCMC3. No significant difference (*p* < 0.05) was found between samples BFP, BXG1, and BXG2, or between samples BXG3, BCMC1, BCMC2, and BCMC3. The high pH value of sample CSD can be attributed to its low TA value.

Total soluble solid (TSS) is a measure of soluble sugars in juices. The TSS of the samples was in the range of 7.40–12.40°Brix (Table 1). Adding the hydrocolloids affected the Brix values of the BFP drink samples. Similar results were reported by Adedokun et al. [36]. The lower TSS in the BFP-coded sample could be attributed to the particular BFP used in this study having a reduced sugar content. However, the addition of the hydrocolloids increased the sugar level across all the samples, BXG1, BXG2, BXG3, BCMC1, BCMC2, and BCMC3, which were not significantly different from one another (*p* < 0.05). Sample CSD has the highest Brix (12.40°Brix), which may be due to the added sugar (sucrose) and synthetic sweetening agents (viz., acesulfame-K and aspartame) included in its formulation.

The viscosity values of the BFP drink samples were in the range of 20.85–30.80 cps. As expected, the addition of a hydrocolloid significantly increased (*p* < 0.05) the viscosity of samples BXG1, BXG2, BXG3, BCMC1, BCMC2, and BCMC3. The least viscous of the samples was the BFP-coded sample, which may be due to the intrinsic rapid sedimentation of BFP in normal solutions. The significant increase in the viscosity of the thickened BFP drink samples was highly appreciated; however, the viscosity value of BXG1 was almost similar to the adopted comparison CSD sample and has no significant difference (*p* < 0.05). Several studies have confirmed that increasing the mass fraction of hydrocolloids enhances the viscosity of fruit-based beverages. For instance, Akkarachaneeyakorn and Tinrat [51] reported that the viscosity of mulberry juice increased proportionally with the addition of XG and CMC. Similarly, Boonditsataporn and Vatthanakul [52] observed that increasing the proportion of XG while reducing the proportion of CMC significantly improved the apparent viscosity and other desirable quality attributes of passion fruit topping sauce.

Table 1 also shows the vitamin C content of the drink samples in the range of 681–800.50 mg/L with the BFP-coded sample having the highest concentration (800.50 mg/L). This value varies with 630 mg/L reported by Stadlmayr et al. [53] and 466 mg/L by Tembo et al. [17]. Variation in the composition of plant-based foods can be attributed to multiple factors, including plant species, environmental conditions, stage of ripeness at harvest, postharvest handling, storage conditions, and the analytical methods employed [17,54]. The significantly high (*p* < 0.05) vitamin C content across the samples could be attributed to the high concentration of ascorbic acids inherent in BFP, which has been reported to be a good source of ascorbic acid [18,55]. Consequently, a 250 mL serving of the formulated BFP drinks delivers ca. 170–200 mg of ascorbic acid, significantly exceeding the daily recommended allowance of 65–90 mg/day for adults [56] and 15–100 mg/day across all age groups of people [56]. This high concentration not only ensures sufficient daily intake, but also enhances the drink’s health benefits, including boosting immune function and providing potent antioxidant protection, thereby reducing the risk of non-communicable diseases, as reported by Carr et al. [57], while remaining well within the safe upper-intake level of 2000 mg/day [58].

### 3.4. Sensory Scores of the BFP Drink Samples Immediately After Formulation and After 2 Weeks of Refrigeration at 4 °C

The results of the preliminary sensory acceptability test of the samples after formulation and after 2 weeks of refrigeration at 4 °C are displayed in Figure 5. (A) Color; (B) taste; (C) mouthfeel; (D) flavor; and (E) general acceptability. The formulated BFP drinks were compared with control (BFP and CSD) samples. The test revealed that the differences in color, mouthfeel, flavor, and general acceptability attributes between the CSD and BXG1 samples were not significant (*p* < 0.05). Sample BXG1 underwent no changes throughout the study period (2 weeks) and no significant difference (*p* < 0.05) was observed. This sample also had the highest mean scores of 6.62 and 6.90 for “like very much” and “general acceptability”, respectively, throughout the study period. Panelists also preferred the “mouthfeel” of sample BXG1 more than that of CSD. The majority of panelists categorically disliked BCMC1, BCMC2, and BCMC3, and scored the drinks very low, based on mouthfeel, taste, flavor, and especially color. Moreover, the color of the BCMC1, BCMC2, and BCMC3 samples was divided into two bands, pale at the top, and dark at the bottom (Figure 6A,B), which can be attributed to the inability of CMC to homogenize well with the BFP and, hence, the inability to control natural sedimentation. Gas formation is also visible in BCMC1 and BCMC2 samples thickened with CMC (Figure 6A,B). These results are in accordance with the recent studies that used various amounts of hydrocolloids, such as the study of Kim et al. that uses XG (1–3%) in several fruit juice formulations [59]. Genovese and Lozano noted that a minimum concentration of XG is necessary to stabilize turbidity in cloudy apple juice, as sensory evaluations revealed consumer preference for samples containing XG [60]. Similarly, in carrot juice, at least 3 g L⁻^1^ of XG is required to maintain cloud stability [61]. However, excessive viscosity may suppress aroma release, potentially reducing flavor intensity and consumer acceptance [62]. Thus, while hydrocolloids improve techno-functional properties, their concentration must be carefully optimized. Gössinger et al.’s study also highlighted the negative impact of excessive XG on flavor typicity and overall sensory appeal [61].

These, along with the high, preliminary sensory acceptance of sample BXG1 (95% BFP: 5% XG), suggest that XG contributes positively to texture and flavor perception and can be effectively utilized in developing a nutrient-rich, functional beverage that aligns with consumer preferences and promotes human health. It provides valuable early-stage insights into the potential consumer appeal of BFP formulations. As emphasized by Stone and Sidel [63], such preliminary sensory screening is a valid and essential step in iterative product development. Moreover, these promising results warrant further validation through broader consumer-based studies, as recommended by the international sensory evaluation standards (ISO 11136:2014) [64,65,66].

### 3.5. Microbial Growth in the BFP Drink

Microbiological analyses of the studied samples displayed the general microbial growth patterns observed in Table 2. Aerobic plate counts (APCs) showed either very low or undetectable microbial counts immediately after preparation in samples BXG1, BXG2, BXG3, and CSD. Also, low microbial counts were detected in BCMC1 (9 CFU/mL), BCMC2 (5 CFU/mL), BCMC3 (2 CFU/mL), and BFP (15 CFU/mL) immediately after preparation. After 2 weeks of constant refrigeration at 4 °C, APCs increased slightly in samples BXG1 (9 CFU/mL), BXG2 (19 CFU/mL), and BXG3 (12 CFU/mL), and significantly in BFP (2.7 log CFU/mL). It is important to note that the limit of detection (LOD) for this study was 1 CFU/mL, as 1 mL of each sample was plated. Thus, no detection equates to <1 CFU/mL, reinforcing the low microbial growth in samples BXG1, BXG2, BXG3, and CSD throughout the study period. Similar trends were noted by Hossain et al., who reported a low initial viable count in a pineapple:papaya:banana:orange juice blend that increased slightly during storage but remained within acceptable limits [67]. Additionally, Begam et al. documented significant initial contamination in a mixed fruit juice made from mango, orange, and pineapple, with a variety of bacterial and mold species [68].

Elaborately, the microbial counts in samples containing XG (BXG1, BXG2, and BXG3) were very low for up to 14 days post-preparation, with microbial levels below safety thresholds, showcasing that XG might provide a synergistic antimicrobial effect in a multi-hurdle formulation strategy. Some studies reported XG demonstrating antimicrobial behavior, especially when used in a formulation, composite films, or in synergy with active agents [69,70]. In contrast, samples BCMC1, BCMC2, and BCMC3 show no significant difference (*p* < 0.05) with the BFP (control) sample. The higher APCs observed in these samples, though within acceptable limits, could be attributed to several factors, such as inherent spoilage microorganisms; environmental conditions during preparation or storage, including the inability of the added CMC to prevent phase separation, which induces spoilage; or the nutrient-rich composition of these samples. Furthermore, CMC, being primarily inert, does not possess antimicrobial properties as reported by Pegg [71], and this may have favor the high microbial growth.

While all the APCs remained below the generally accepted safety threshold of 10^3^ CFU/mL for ready-to-drink beverages, as outlined by the International Commission on Microbiological Specifications for Foods (ICMSF) [72], it is important to recognize that these results alone do not necessarily confirm the microbiological safety of the products. A specific analytical evaluation of foodborne bacterial pathogens (e.g., *Salmonella*, *Escherichia coli*, *Listeria monocytogenes*, etc.) was not conducted. Therefore, while the APCs were low enough not to impact microbial product quality within the study period (2 weeks), the ultimate safety of these beverages would need to be determined in a more detailed bacterial pathogen challenge study. However, microbial counts, especially in the control (BFP-coded) sample, continued to rise during the storage period, suggesting that growth could continue if storage was prolonged. This highlights the importance of conducting further accelerated and comprehensive storage stability studies, as microbial counts are likely to increase over time, even under refrigerated conditions (4 °C). Also, the three-times storage trials at room temperature displayed a rapid increase in microbial counts, resulting in too-numerous-to-count (TNTC) levels by ≤day 4 across all the samples, excluding CSD, which was why those trials were deemed to have failed due to high contamination.

## 4. Conclusions

This study underscores the nutritional and functional potential of BFP drinks, particularly for their high energy content (330.64 kcal/100 g) and exceptional vitamin C concentration, with a 250 mL serving providing 170–200 mg, well above the daily recommended intake for adults. Among the BFP-based formulations, a preliminary acceptability test using 7-point sensory hedonic scale indicated that sample BXG1 (95% BFP, 5% XG) was preferred over a commercial synthetic beverage, supporting the formulation’s sensory viability. Though, further studies involving large-scale consumer panels are recommended to validate these findings across diverse demographic groups and consumption contexts. However, room-temperature storage (ca. 25 ± 2 °C) resulted in significant microbial proliferation and reached TNTC level, rendering the drink unsuitable for consumption within four days. Refrigerated storage (4 °C) effectively preserved the sensory and microbial qualities for up to 14 days. These findings emphasize the importance of hydrocolloids in improving the sensory properties of BFP drinks and highlight the need for advanced preservation strategies, such as the incorporation of natural preservatives, to extend shelf life. Future research should explore preservation techniques to ensure the safety and viability of BFP drinks at room temperature, especially for regions with limited access to balanced nutrition.

## Figures and Tables

**Figure 1 polymers-17-01396-f001:**
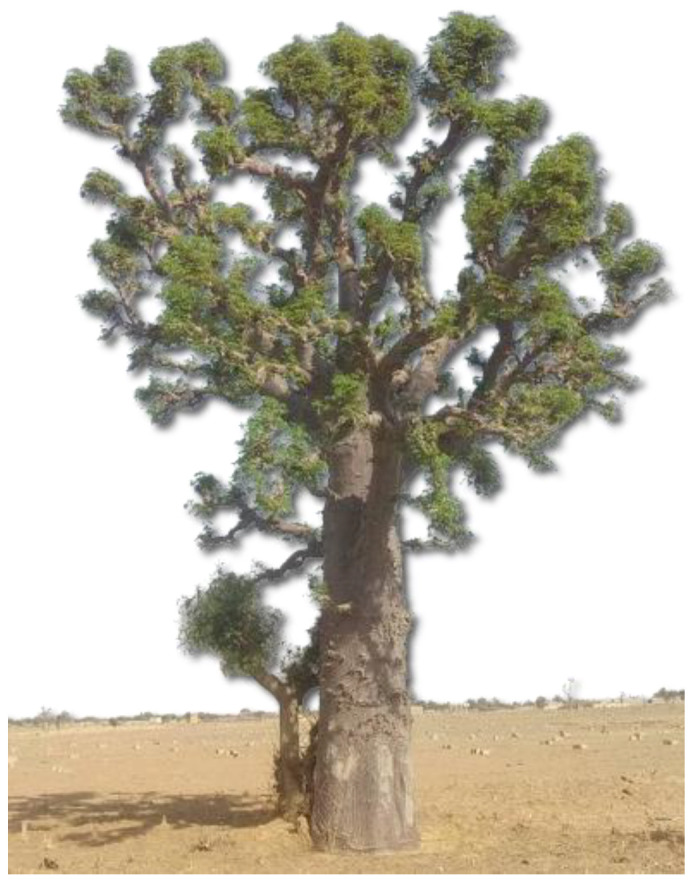
Baobab (*Adansonia digitata* L.) tree.

**Figure 2 polymers-17-01396-f002:**
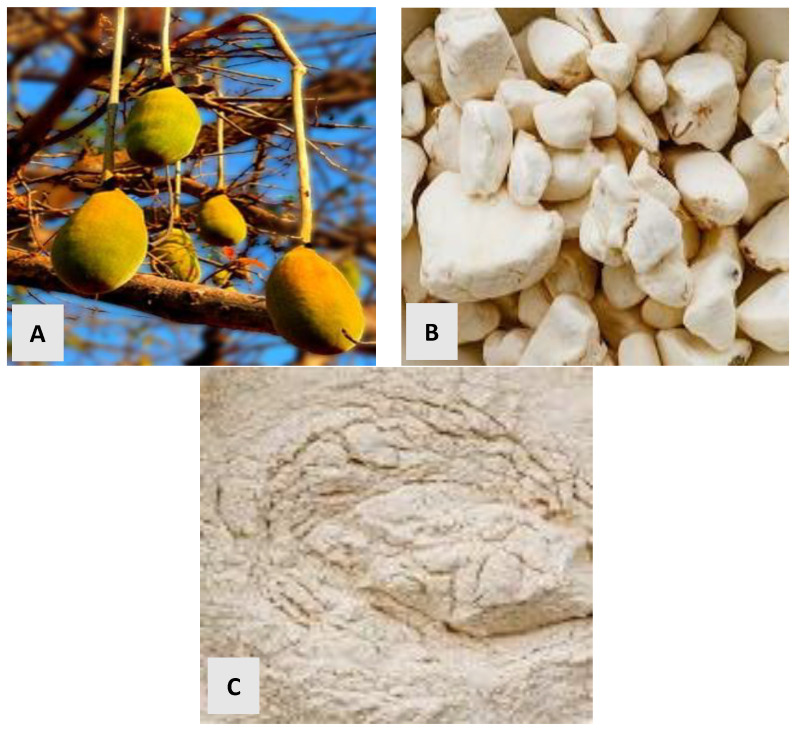
(**A**) Baobab fruit capsules; (**B**) baobab fruit pulp granules; (**C**) baobab fruit pulp powder.

**Figure 3 polymers-17-01396-f003:**
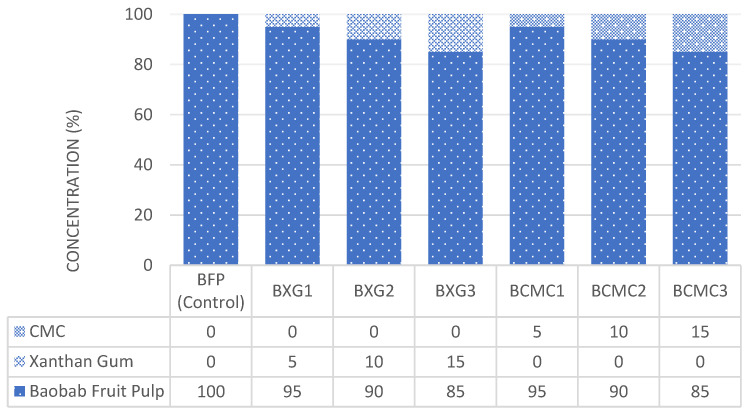
BFP drinks formulation.

**Figure 4 polymers-17-01396-f004:**
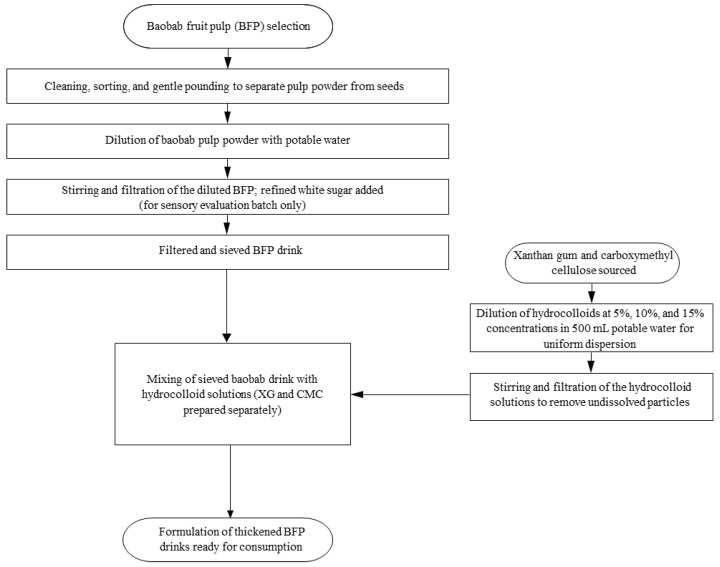
Flow diagram for the production of enhanced BFP drinks.

**Figure 5 polymers-17-01396-f005:**
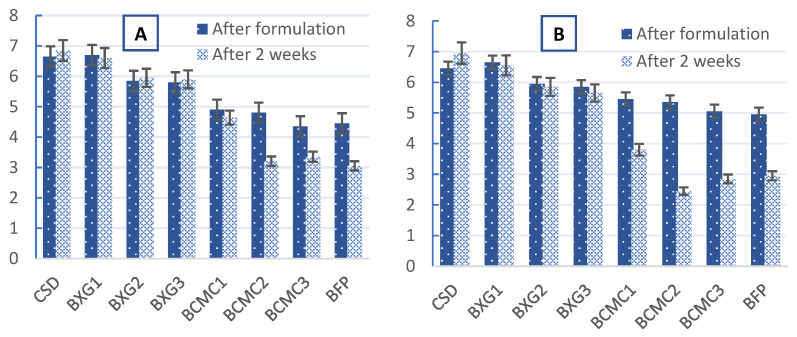
Sensory scores of the formulated BFP drinks and control samples. (**A**) Color; (**B**) taste; (**C**) mouthfeel; (**D**) flavor; and (**E**) general acceptability.

**Figure 6 polymers-17-01396-f006:**
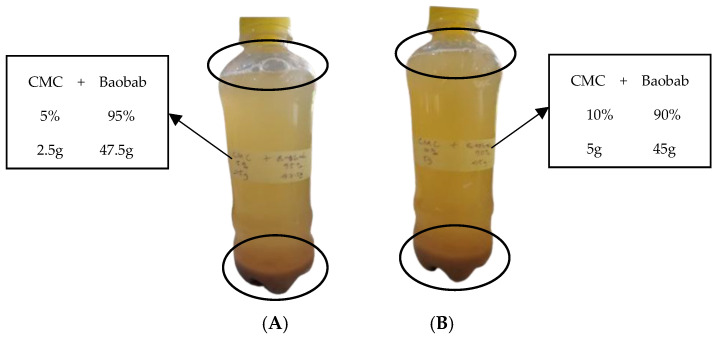
Representative samples: (**A**) BCMC1 and (**B**) BCMC2.

**Table 1 polymers-17-01396-t001:** Results of physicochemical properties and vitamin C value.

Parameters ^2^					
Sample Codes ^1^	pH	TA (g Ascorbic Acid/100 mL)	TSS (°Brix)	Viscosity (cps) at 30 °C	Vitamin C (mg/L)
BXG1	3.7 ± 0.1 ^C^	4.1 ± 0.1 ^A^	8.3 ± 0.2 ^BC^	25.5 ± 0.2 ^A^	681 ± 1.0 ^C^
BXG2	3.5 ± 0.1 ^B^	4.1 ± 0.5 ^A^	7.6 ± 0.9 ^BC^	27.2 ± 0.4 ^C^	755 ± 35.4 ^B^
BXG3	4.0 ± 0.1 ^B^	3.8 ± 1.1 ^B^	8.5 ± 0.5 ^B^	30.8 ± 0.9 ^D^	653 ± 0.0 ^C^
BCMC1	3.6 ± 0.1 ^BC^	3.7 ± 0.3 ^B^	9.1 ± 0.1 ^B^	26.3 ± 0.1 ^B^	560 ± 0.0 ^D^
BCMC2	3.6 ± 0.0 ^C^	3.9 ± 0.1 ^B^	9.0 ± 0.0 ^B^	28.0 ± 0.1 ^C^	560 ± 0.0 ^D^
BCMC3	3.6 ± 0.1 ^C^	3.9 ± 1.0 ^B^	8.5 ± 0.5 ^C^	29.8 ± 0.3 ^D^	656 ± 5.7 ^C^
BFP	3.10 ± 0.1 ^A^	4.2 ± 0.0 ^A^	7.4 ± 0.3 ^D^	20.9 ± 0.9 ^E^	801 ± 11.0 ^A^
CSD	4.2 ± 0.1 ^D^	2.7 ± 0.0 ^C^	12.4 ± 0.1 ^A^	24.05 ± 0.21 ^A^	290 ± 14.1 ^E^

^1^ BXG1 (95% baobab fruit pulp and 5% xanthan gum); BXG2 (90% baobab fruit pulp and 10% xanthan gum); BXG3 (85% baobab fruit pulp and 15% xanthan gum); BCMC1 (95% baobab fruit pulp and 5% CMC); BCMC2 (90% baobab fruit pulp and 10% CMC); BCMC3 (85% baobab fruit pulp and 15% CMC); BFP (100% baobab fruit pulp—control); and CSD (synthetic commercial drink—control). ^2^ Each value in the table represents the mean of triplicate ± standard deviation. Obtained values in the same column followed by different superscript letters are significantly different at *p* < 0.05.

**Table 2 polymers-17-01396-t002:** Results of microbial growth of the samples studied.

Sample Codes ^1^	Total Plate Count (TPC) (cfu/mL)	
	Immediately After Formulation	After 2 Weeks
BXG1	No Growth	0.9 × 10^1^
BXG2	No Growth	1.9 × 10^1^
BXG3	No Growth	1.2 × 10^1^
BCMC1	0.9 × 10^1^	5.2 × 10^2^
BCMC2	0.5 × 10^1^	3.6 × 10^2^
BCMC3	0.2 × 10^1^	4.2 × 10^2^
BFP	1.5 × 10^1^	5.6 × 10^2^
CSD	No Growth	3.6 × 10^1^

^1^ BXG1 (95% baobab fruit pulp and 5% xanthan gum); BXG2 (90% baobab fruit pulp and 10% xanthan gum); BXG3 (85% baobab fruit pulp and 15% xanthan gum); BCMC1 (95% baobab fruit pulp and 5% CMC); BCMC2 (90% baobab fruit pulp and 10% CMC); BCMC3 (85% baobab fruit pulp and 15% CMC); BFP (100% baobab fruit pulp); and CSD (synthetic commercial drink—control). Each value in the table represents the mean of triplicate ± standard deviation.

## Data Availability

Data will be made available on request.

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
