# Peer review of "Enhancing the Sensory Quality, Stability, and Shelf Life of Baobab Fruit Pulp Drinks: The Role of Hydrocolloids"

_polymers, 2025, doi:10.3390/polym17101396_

Round 1

Reviewer 1 Report

Comments and Suggestions for Authors

The manuscript "Enhancing Sensory Quality, Stability, and Shelf Life of Baobab Fruit Pulp Drinks: The Role of Hydrocolloids" is well-written and presents a well-conducted study on the use of xanthan gum (XG) and carboxymethyl cellulose (CMC) as thickening and stabilizing agents for Baobab fruit pulp drinks. The research is methodically structured and contributes valuable insights into improving the sensory quality and shelf life of such beverages.

However, the following minor revisions are required:

  1. Reference Citations: The journal Polymer follows a number format for references. Please adjust the citation style accordingly.
  2. Figures: No figures were found in the manuscript, only figure legends. Please ensure that the figures are uploaded correctly.
  3. Line 93: Remove the comma after the name Nour.
  4. Introduction Section: The description of XG and CMC is minimal. Please elaborate on their functional roles, safety aspects, and prior applications as thickening and stabilizing agents. Cite relevant studies where these hydrocolloids have been used in similar food matrices.
  5. Line 114: The "C" in "chemical" should be capitalized.
  6. Reference Section: Remove underlines from DOIs in the reference list as per journal formatting guidelines.

Addressing these revisions will improve the clarity, completeness, and adherence to journal formatting requirements.

Author Response

Reviewer’s Comment and Suggestion for Author

The manuscript “Enhancing Sensory Quality, Stability, and Shelf Life of Baobab Fruit Pulp Drinks: The Role of Hydrocolloids” is well-written and presents a well-conducted study on the use of xanthan gum (XG) and carboxymethyl cellulose (CMC) as thickening and stabilizing agents for Baobab fruit pulp drinks. The research is methodically structured and contributes valuable insights into improving the sensory quality and shelf life of such beverages. However, the following minor revisions are required.

Response: We thank the reviewer for finding our work interesting and valuable.

Minor revisions:

  1. Reference Citations: The journal Polymer follows a number format for references. Please adjust the citation style accordingly.

Response: We thank the reviewer for this insightful correction. We have formatted the citation style to reflect MDPI citation style throughout the entire manuscript accordingly.

  1. Figures: No figures were found in the manuscript, only figure legends. Please ensure that the figures are uploaded correctly.

Response: We thank you for this observation. However, you may wish to note that earlier during the submission of this manuscript, we mistakenly submitted the figures in a separate file, but now all figures have now been provided at their respective position in the manuscript and in accordance with the MDPI’s instruction for author. 

  1. Line 93: Remove the comma after the name Nour.

Response: The comma after the name Nour has been removed and all other citation-related corrections have been similarly corrected.

  1. Introduction Section: The description of XG and CMC is minimal. Please elaborate on their functional roles, safety aspects, and prior applications as thickening and stabilizing agents. Cite relevant studies where these hydrocolloids have been used in similar food matrices.

Response:  We thank you for bringing our attention unto this. We have elaborated on the several functionalities of XG and CMC as enhancements agent. Relevant studies where these hydrocolloids have been used in different food matrices has been cited (line 103-118).

  1. Line 114: The "C" in "chemical" should be capitalized.

Response: “chemical” has been changed to “Chemical” as suggested (line 135).

  1. Reference Section: Remove underlines from DOIs in the reference list as per journal formatting guidelines

Response: the underlined DOIs have been corrected and the entire references list section has been formatted to the MDPI standard accordingly.

Final Comment:

Addressing these revisions will improve the clarity, completeness, and adherence to journal formatting requirements.

Response: We thank you for all your valuable suggestions. We have tried to address them as adequately as possible.

Reviewer 2 Report

Comments and Suggestions for Authors

The technical content of this manuscript is quite poor and consequently it can have only limited impact in the technology for baobab fruit pulp (BFP) processing. In addition, this manuscript contains several criticisms to be solved too.

- The evaluation of the XG and CMC bacterial inhibiting effect for the baobab fruit pulp is probably the only experimental result of some importance contained in this manuscript. However, also such experimental result seems to be quite poor and a deeper investigation is required. Indeed, the antiseptic behaviour of both XG and CMC hydrocolloids does not seem to be considerable. The antiseptic behaviour observed immediately after the XG and CMC addition is less important than the bacterial inhibiting effect after a prolonged time period since BFP always require some storage time before use. Anyhow, the effect of CMC addition just after the hydrocolloid preparation seems negligible and probably comparable with the associated errors. Differently, the XG addition could be considered as having immediate effect on the aerobic plate count because no bacterial cells growth is observed. The total plate count (TPC) after 2 weeks is a much more important behaviour for evaluating the stabilizing capability of these additives on BFP. About CMC addition after a time lapse of two weeks, there is not antiseptic effect; indeed, the observed TCP of 5.6x102 without CMC and 5.2x102 with 5% of CMC are similar results. Even at high CMC percentage (15%), the effect of CMC is poor (i.e., 4.2x102 cells instead of 5.6x102 cells). In addition, such results seem to be fluctuating with the XG and CMC percentage.

- According to the error values associated with numbers given in the three tables, these numbers should be rounded to the first decimal place and the errors should also be expressed with a single significant number.

- The duplicates of tables placed at the end of the manuscript should be removed.

- If histograms are built by using data given in the three Tables, a few graphs could be included in this manuscript, which lacks of images/schemes/photos, making the results more clear to understand.

Author Response

Reviewer’s Comment and Suggestion for Author

The technical content of this manuscript is quite poor and consequently it can have only limited impact in the technology for baobab fruit pulp (BFP) processing. In addition, this manuscript contains several criticisms to be solved too.

Response: We sincerely thank the reviewer for the critical assessment of our manuscript. We deeply value your input and understand that strong scrutiny is essential to ensuring scientific rigor. We also appreciate the opportunity to clarify certain aspects of the study, particularly in relation to the observed microbial behavior and the primary aims of our work.

Comments

  1. The evaluation of the XG and CMC bacterial inhibiting effect for the baobab fruit pulp is probably the only experimental result of some importance contained in this manuscript. However, also such experimental result seems to be quite poor and a deeper investigation is required. Indeed, the antiseptic behaviour of both XG and CMC hydrocolloids does not seem to be considerable. The antiseptic behaviour observed immediately after the XG and CMC addition is less important than the bacterial inhibiting effect after a prolonged time period since BFP always require some storage time before use. Anyhow, the effect of CMC addition just after the hydrocolloid preparation seems negligible and probably comparable with the associated errors. Differently, the XG addition could be considered as having immediate effect on the aerobic plate count because no bacterial cells growth is observed. The total plate count (TPC) after 2 weeks is a much more important behaviour for evaluating the stabilizing capability of these additives on BFP. About CMC addition after a time lapse of two weeks, there is not antiseptic effect; indeed, the observed TCP of 5.6x102 without CMC and 5.2x102 with 5% of CMC are similar results. Even at high CMC percentage (15%), the effect of CMC is poor (i.e., 4.2x102 cells instead of 5.6x102 cells). In addition, such results seem to be fluctuating with the XG and CMC percentage.

Response:

The central objective of our study was to explore the functional role of these hydrocolloids, xanthan gum (XG) and carboxymethyl cellulose (CMC) in improving physicochemical stability, rheological behavior, sensory acceptability, and storage effects, with each contributing critical insights into formulation science of baobab fruit pulp (BFP) beverages. Given the natural susceptibility of BFP to microbial spoilage, we included microbial analyses primarily to assess whether these hydrocolloids, known to alter water activity and molecular mobility, might also contribute to microbial inhibition as part of a broader hurdle approach. This was especially relevant as we sought to determine the feasibility of hydrocolloid use in minimally processed beverages that might be stored under refrigeration for short durations. In line with this scope, our microbial evaluation was designed to track aerobic plate count (APC) over a two-week refrigerated storage period (4 °C), not to establish a strong antiseptic efficacy per se.

Moreover, the microbiological results are not presented as definitive antimicrobial claims, nor were they designed as a pathogen challenge study as we clearly mentioned in the manuscript (line 558-564). Our microbial analysis focused on the fundamental total aerobic mesophilic counts as an initial indicator of microbiological quality, has appeared to be successful. The consistently low counts in XG-containing samples versus significantly higher counts in the control and CMC samples indicate the clear difference in their individualistic activity towards BFP-based drinks’ shelf-life preservation. This has been clearly mentioned in the manuscript that among the possible justification of CMC, might be its inability to control the phase separation in BFP drinks formulation. Additionally, Pegg, (2012) study have reported that CMC, being primarily inert, does not possess “good” antimicrobial properties. We have already acknowledged that CMC did not exhibit statistical difference on a robust antimicrobial activity across all concentrations and this aligns with the known behavior of CMC as a passive stabilizer rather than an antimicrobial agent. In addition, this had supported our conclusion that CMC doesn’t contributes more to any of physical stability (e.g., sedimentation control, viscosity enhancement) as well as microbial stabilization. In contrast, and to elaborate on the consistent suppression of APC growth in XG-treated samples, from day 0 through day 14, that might have likely arisen from a combination of factors: the intrinsic viscosity barrier created by XG and possible synergistic effects that hinder bacterial proliferation. Similarly, XG has demonstrated antimicrobial behavior under certain conditions in other formulations, especially when used as part of composite films or in synergy with active agents as reported by Gössinger et al., (2018), Huang et al., (2025), and Kim et al., (2017). This emergent behavior deserves further study, and our findings form a credible platform for future work to isolate and characterize these effects under controlled bacterial inoculation. These findings revealed meaningful differences in formulation science and suggest that XG contributes to a more microbially stable environment in BFP beverages under refrigerated storage. That said, our study is scientifically meaningful and merits further exploration. These results are not negligible, rather they point toward a possible stabilizing or inhibitory matrix effect of XG in acidic, polyphenol-rich environments like BFP. Importantly, these observations illustrated that hydrocolloids may offer “complementary benefits” in a multi-hurdle formulation strategy.

On variability in microbial counts across hydrocolloid concentrations, likely results from subtle batch-to-batch differences, oxygen exposure, and nutrient diffusion behavior as viscosity changes, especially in the CMC groups. These are normal fluctuations within a biological matrix and do not indicate random noise or experimental error. Importantly, the overall microbial trend is clear, XG suppresses growth consistently, while CMC does not. This conclusion is drawn from statistically replicated trials (n=3, each in duplicate) and supported by the broader literature on hydrocolloid behavior in acidic beverage systems. Indeed, the TPC of 5.6×102 in the control sample and that of all the CMC containing samples (including the BCMC1 with 5.2×102 CFU/mL) showed no statistical differences , and our study has shown that. We elaborately interpret this finding within the exhibited behavior of CMC and accurately reflect this limitation in the discussion section (Line 550-554). The results reinforce the point that CMC should not be relied upon for microbial control in acidic fruit-based beverages unless combined with other preservation strategies.

We would like to reiterate that we have already recommended that, future studies should explore dedicated bacterial challenge trials, using model organisms such as E. coli, L. monocytogenes, and S. aureus, to delineate the microbial stabilization mechanisms at play. These trials should be conducted under aseptic and controlled inoculation conditions, and ideally span both refrigerated and ambient storage periods. Such studies are vital to establish a defensible microbiological safety profile for baobab-based beverages but are beyond the intended scope of the current formulation-centered study. We fully acknowledge (the revised manuscript - line 534-551) that deeper investigations, including strain-specific challenge tests under controlled contamination scenarios is necessary, to elucidate the full microbiological profile. Yet, those belong to a separate stand-alone study and outside the scope of our present study. We explicitly determined (among our future study directions) to conduct an intensive microbiologically study with a dedicated scope towards involving natural antimicrobials and bio-preservation.

  1. Comment

According to the error values associated with numbers given in the three tables, these numbers should be rounded to the first decimal place and the errors should also be expressed with a single significant number.

Response:

We thank the reviewer for this valuable addition. Values displayed in the tables are now rounded to the first decimal place and the standard deviations are now expressed in a single significant number as suggested.

  1. Comment

 The duplicates of tables placed at the end of the manuscript should be removed

Response:

The duplicated tables are being removed accordingly.

  1. Comment

If histograms are built by using data given in the three Tables, a few graphs could be included in this manuscript, which lacks of images/schemes/photos, making the results more clear to understand

Response:

We thank the reviewer for this valuable insight. We took into account to convert the Table 2. A&B (Sensory Scores of the Studied Samples after formulation and after 2 Weeks of Storage at 4°C) and now became figure 5. (Sensory Scores of the formulated BFP Drink and control Samples. (A) Color; (B) Taste; (C) Mouthfeel; (D) Flavor; and (E) General Acceptability). Now all the figures, images and a formulation flow diagram have now been positioned back to their respective places.

New references added to the manuscript  

Gössinger, M., Buchmayer, S., Greil, A., Griesbacher, S., Kainz, E., Ledinegg, M., Leitner, M., Mantler, A., Hanz, K., & Bauer, R. (2018). Effect of xanthan gum on typicity and flavour intensity of cloudy apple juice. Journal of Food Processing and Preservation, 42(10), e13737.

Huang, S., An, S., Kannan, P. R., Wahab, A., Ali, S., Xiaoqing, L., Suhail, M., Iqbal, M. Z., & Kong, X. (2025). Development and characterization of biodegradable antibacterial hydrogels of xanthan gum for controlled ciprofloxacin release. International Journal of Biological Macromolecules, 142637.

Kim, H., Hwang, H., Song, K., & Lee, J. (2017). Sensory and rheological characteristics of thickened liquids differing concentrations of a xanthan gum‐based thickener. Journal of Texture Studies, 48(6), 571–585.

Pegg, A. M. (2012). The application of natural hydrocolloids to foods and beverages. In Natural food additives, ingredients and flavourings (pp. 175–196). Elsevier.

Reviewer 3 Report

Comments and Suggestions for Authors

The paper “Enhancing Sensory Quality, Stability and Shelf Life of Baobab Fruit Pulp Drinks: The Role of Hydrocolloids” was focused on the investigation of the effects of two hydrocolloids, xanthan gum and carboxymethyl cellulose on the viscosity, shelf-life stability, and consumer acceptance of baobab fruit pulp drinks.

The aims of the study are expressed.

The experimental program is described in such manner that it can be applied.

The obtained results are concise, presented and discussed.

Conclusions are drawn according to the obtained data.

Figures mentioned in the manuscript were not available for the review process. At the end of the paper there is a list of figure captions but it is followed by Tables instead of Figures.

The authors will find bellow some corrections and adjustments that should be addressed.

Keywords

  • “CMC” should be replaced with the entire name of the hydrocolloid.

  1. Introduction
  • A careful read off the manuscript is required in order to revise the figure numbering. They should be numbered in the order of their appearance in the text. At present, the first figure is Figure 3A.

2.4.1. Titratable Acidity (TA) and pH Measurements

  • Please revise the equation 1 and the explanation for its constituents since they are not in agreement.

2.4.4. Water Absorption Capacity (WAC)

  • Information for all the equipment and set parameters should be provided (g. information about the equipment used for centrifugation is missing).

2.7. Proximate Composition

  • The exposed methods should be completed with the equations used to establish the moisture content, ash content etc.

3.0. Results and Discussion

  • It is recommended to improve the section by adding a comparison of the obtained results with those reported elsewhere (if available) for using other types of hydrocolloids.
  • The methodology was developed at laboratory level. Is there a possibility to realize a scale up in order to highlight the viability of the proposed process? With what limitations?

Author Response

Reviewer’s Comment and Suggestion for Author

The paper “Enhancing Sensory Quality, Stability and Shelf Life of Baobab Fruit Pulp Drinks: The Role of Hydrocolloids” was focused on the investigation of the effects of two hydrocolloids, xanthan gum and carboxymethyl cellulose on the viscosity, shelf-life stability, and consumer acceptance of baobab fruit pulp drinks. The aims of the study are expressed. The experimental program is described in such manner that it can be applied. The obtained results are concise, presented and discussed. Conclusions are drawn according to the obtained data.

Response: We are thankful to the reviewer for finding our work concise, expressive and applicable.

Corrections and adjustments:

  1. Figures: Figures mentioned in the manuscript were not available for the review process. At the end of the paper there is a list of figure captions but it is followed by Tables instead of Figures.

Response: Please note that all figures are now available within the manuscript accordingly, kindly check them one more time.

  1. Keyword: “CMC” should be replaced with the entire name of the hydrocolloid.

Response: CMC have been rewritten in its full name as suggested (line 31).

  1. Keyword: A careful read off the manuscript is required in order to revise the figure numbering. They should be numbered in the order of their appearance in the text. At present, the first figure is Figure 3A.

Response: Thank you for your insightful comment. The manuscript has been revised and all the figures have been corrected, numbered and placed accordingly.

  1. Titratable Acidity (TA) and pH Measurements: Please revise the equation 1 and the explanation for its constituents since they are not in agreement.

Response: equation 1 has been revised and the explanation of its constituents are now in agreement with the formula in accordance to the AOAC 942.15 titrimetric method (line 177-187).

  1. Water Absorption Capacity (WAC): Information for all the equipment and set parameters should be provided (g. information about the equipment used for centrifugation is missing).

Response: Thank you for the suggestion. The model, brand, name and location of the manufacturer have been added as suggested (line 191-192 & 230-232). All other equipment used are disclosed in the manuscript.

  1. Results and Discussion: It is recommended to improve the section by adding a comparison of the obtained results with those reported elsewhere (if available) for using other types of hydrocolloids.

Response:  We thank you for your valuable recommendation of adding similar studies on hydrocolloids. Please note that, we have added several studies that have utilized hydrocolloids to enhance the tecno-functional properties of several fruit-based beverages in order to improve our discussion as you suggested (line 489-498).

  1. The methodology was developed at laboratory level. Is there a possibility to realize a scale up in order to highlight the viability of the proposed process? With what limitations?

Response: Thank you for this forward-looking comment. We are pleased to note that your observation aligns with the long-term vision of our study. Although the methodology was implemented at a laboratory scale, it was strategically designed to simulate real-world formulation and processing dynamics for Baobab Fruit Pulp (BFP) beverages, particularly addressing challenges associated with viscosity, sensory quality, and microbial stability under tropical conditions. The integration of these hydrocolloids was guided by scalable formulation principles, and our findings demonstrate clear potential for industrial adaptation. Specifically, the BXG1 formulation (95% BFP: 5% Xanthan Gum) not only achieved superior sensory acceptability but also exhibited promising shelf stability under refrigeration, pointing to a formulation platform that could benefit from further enhancement via natural preservation systems, which we have already finds some answers to that. Moreover, we are currently preparing for a pilot-scale study that will explore batch-level processing, industrial mixing protocols, and cold-chain logistics under field-relevant conditions. We are encouraged by the shared interest in advancing this work beyond the laboratory and believe that these developments will contribute significantly to sustainable food innovation, functional beverage development, and the valorization of underutilized yet nutritious indigenous fruits.

Round 2

Reviewer 3 Report

Comments and Suggestions for Authors

The paper “Enhancing Sensory Quality, Stability and Shelf Life of Baobab Fruit Pulp Drinks: The Role of Hydrocolloids” was focused on the investigation of the effects of two hydrocolloids, xanthan gum and carboxymethyl cellulose on the viscosity, shelf-life stability, and consumer acceptance of baobab fruit pulp drinks.

I congratulate the authors for taking into account the suggestions of the reviewers and for their efforts in bringing the necessary clarifications and making the requested changes. The work is much improved compared to the initial version and can be accepted for publication.

I congratulate the authors for taking into account the suggestions of the reviewers and for their efforts in bringing the necessary clarifications and making the requested changes. The work is much improved compared to the initial version and can be accepted for publication.

Author Response

Reviewer comments.

The paper “Enhancing Sensory Quality, Stability and Shelf Life of Baobab Fruit Pulp Drinks: The Role of Hydrocolloids” was focused on the investigation of the effects of two hydrocolloids, xanthan gum and carboxymethyl cellulose on the viscosity, shelf-life stability, and consumer acceptance of baobab fruit pulp drinks.

I congratulate the authors for taking into account the suggestions of the reviewers and for their efforts in bringing the necessary clarifications and making the requested changes. The work is much improved compared to the initial version and can be accepted for publication.

Response. We are encouraged with the feedback and the improvements of the manuscript after the first reviewer’s comments.